Rediscovery of Laura’s glassfrog Nymphargus laurae (Anura: Centrolenidae) with new data on its morphology, colouration, phylogenetic position and conservation in Ecuador

http://orcid.org/0000-0003-4626-7926 Sánchez-Carvajal María José 1 2
Reyes-Ortega Grace C. 1 2
http://orcid.org/0000-0002-6132-2738 Cisneros-Heredia Diego F. 3 4
http://orcid.org/0000-0001-9464-3688 Ortega-Andrade H. Mauricio 1 2 4 mauricio.ortega@ikiam.edu.ec
1 Ingeniería en Ecosistemas, Facultad de Ciencias de la Vida, Universidad Regional Amazónica Ikiam , Tena, Napo , Ecuador
2 Grupo de Investigación en Biogeografía y Ecología Espacial, Universidad Regional Amazónica Ikiam , Tena, Napo , Ecuador
3 Colegio de Ciencias Biológicas y Ambientales, Instituto de Biodiversidad Tropical iBIOTROP, Museo de Zoología, Laboratorio de Zoología Terrestre, Universidad San Francisco de Quito , Quito, Pichincha , Ecuador
4 División de Herpetología, Instituto Nacional de Biodiversidad , Quito, Pichincha , Ecuador
Morrone Juan J.
Electronic publication date: 2021 Dec 23
Publication date: 2021
Volume: 9
Electronic Location ID: e12644
Received 2021 Oct 15; Accepted 2021 Nov 26
Copyright: © 2021 Sánchez-Carvajal et al.
Copyright year: 2021
Copyright holder: Sánchez-Carvajal et al.
License: This is an open access article distributed under the terms of the Creative Commons Attribution License, which permits unrestricted use, distribution, reproduction and adaptation in any medium and for any purpose provided that it is properly attributed. For attribution, the original author(s), title, publication source (PeerJ) and either DOI or URL of the article must be cited.
License URL: https://creativecommons.org/licenses/by/4.0/

Keywords: Phylogeny, Taxonomy, Glassfrogs, Colonso-Chalupas biological reserve, Systematics

Funding: Secretaría Nacional de Ciencia y Tecnología del Ecuador PIC-17-BENS-001 World Academy of Sciences 16-095 Smithsonian Institution World Wildlife Fund Secretaría de Educación Superior, Ciencia, Tecnología e Innovación Universidad San Francisco de Quito Gordon and Betty Moore Foundation This work was supported by the following projects: “On the quest of the golden fleece in Amazonia: The first herpetological DNA—barcoding expedition to unexplored areas on the Napo watershed, Ecuador”, funded by the Secretaría Nacional de Ciencia y Tecnología del Ecuador (Senescyt-ENSAMBLE Grant #PIC-17-BENS-001), and The World Academy of Sciences (TWAS Grant #16-095, granted to H. Mauricio Ortega-Andrade). Research by Diego Francisco Cisneros-Heredia was supported by the Smithsonian Women’s Committee, Smithsonian Institution (2002 Research Training Program, National Museum of Natural History), World Wildlife Fund (WWF-EFN, Russel E. Train Education for Nature Program), Secretaría de Educación Superior, Ciencia, Tecnología e Innovación (SENESCYT, Programa “Becas de Excelencia”), Universidad San Francisco de Quito (Chancellor grants, COCIBA grants, Collaboration grants, projects HUBI ID 34, 36, 39, 48, 1057, 7703, 12253, 13524), and “Proyecto Descubre Napo”, an initiative of Universidad San Francisco de Quito in association with Wildlife Conservation Society and funded by the Gordon and Betty Moore Foundation as part of the project: WCS Consolidating Conservation of Critical Landscapes (mosaics) in the Andes. The funders had no role in study design, data collection and analysis, decision to publish, or preparation of the manuscript.

==============================
We report the rediscovery of Laura’s Glassfrog, Nymphargus laurae Cisneros-Heredia & McDiarmid, 2007, based on two specimens collected at the Colonso-Chalupas Biological Reserve, province of Napo, Ecuador. The species was described and known from a single male specimen collected in 1955 at Loreto, north-eastern Andean foothills of Ecuador. Limited information was available about the colouration, systematics, ecology, and biogeography of N. laurae. We provide new data on the external morphology, colouration, distribution and comment on its conservation status and extinction risk. We discuss the phylogenetic relationships of N. laurae, which forms a clade together with N. siren and N. humboldti. The importance of research in unexplored areas must be a national priority to document the biodiversity associated, especially in protected areas.

Introduction

Nymphargus Cisneros-Heredia & McDiarmid, 2007 currently includes 42 described species of glassfrogs (family Centrolenidae), and 21 of them occur in Ecuador (Guayasamin et al., 2020; Frost, 2021). Despite increasing efforts to better understand the diversity, natural history, ecology and distribution of glassfrogs in the tropical Andes, several species of Nymphargus remain known only from their type localities or few collected specimens (e.g., Nymphargus buenaventura (Cisneros-Heredia & Yánez-Muñoz, 2007; Yánez-Muñoz et al., 2014), N. laurae (Cisneros-Heredia & McDiarmid, 2007), N. lindae Guayasamin in Guayasamin et al. (2020), N. manduriacu Guayasamin et al. (2019)).

Laura’s Glassfrog Nymphargus laurae was described based on a male specimen collected in 1955 at Loreto, on the north-eastern foothills of the Andes of Ecuador (Cisneros-Heredia & McDiarmid, 2007). More than 60 years have passed since the collection of the holotype and single known specimen of N. laurae, and no additional individuals or information has become available for the species (Guayasamin et al., 2020). Since its description, numerous herpetologists have searched for N. laurae along the eastern slopes of the Andes of Ecuador without success. Due to its apparent rarity, restricted distribution, and extensive habitat change and loss at the type-locality, N. laurae was classified as Endangered at the national level (Ortega-Andrade et al., 2021) and Critically Endangered at the global level (Cisneros-Heredia, 2008).

Between 2016–2018, we collected two individuals of Nymphargus laurae at the Colonso-Chalupas Biological Reserve. Based in this findings, we provide new information about the morphological and chromatic variation, natural history, conservation status and extinction risk and reveal for the first time the phylogenetic position of N. laurae in Centrolenidae.

Materials and methods

Study area and field surveys

The Colonso-Chalupas Biological Reserve is a national protected area located on the foothills of the north-eastern Andes of Ecuador, in the province of Napo. This biological reserve protects 932.46 km2, extending between 560–4,432 m elevation and being home to a variety of ecosystems, from tropical evergreen forests to paramos (Van der Hoek et al., 2018). It is part of an ecological corridor with two neighbouring protected areas: Antisana Ecological Reserve, to the north, and Llanganates National Park, to the south (Ramis, Álvarez-Solas & Peñuela, 2018; Van der Hoek et al., 2018). Montane evergreen cloud forests in Colonso-Chalupas are characterised by a great variety of trees of the families Melastomataceae, Solanaceae, Myrsinaceae, Aquifoliaceae, Araliaceae, Rubiaceae. Those trees reach up to 15–25 m in height, showing gnarled trunks and branches, and dense and compact crowns covered by epiphytes, including orchids, bromeliads, aroids, and ferns (Ministerio del Ambiente del Ecuador, 2012, 2013). This ecosystem is usually covered by mist, either constantly or during the early morning and late afternoon (Ramis, Álvarez-Solas & Peñuela, 2018). The average annual rainfall is 4,620 mm, and the average annual temperature is 28.7 °C. The rainy season extends between March and July, with 448 mm of monthly average rainfall and 23.5 °C of monthly average temperature. The dry season is between August and January, with 353 mm of average monthly precipitation and 23.9 °C of average monthly temperature (INAMHI, 2015). Nocturnal surveys for collection of amphibians and reptiles were conducted at the Colonso-Chalupas Biological Reserve from 19h00–02h00, at a stream nearby to Ikiam’s Scientific Station, on 17 October 2016 (0.9348°S 77.9270°W, 1,506 m) and at the Narpa stream, on 09 June 2018 (0.9353°S 77.9268°W, 1,440 m) (Fig. 1). Environmental Ministry of Ecuador provided full approval for this research (MAE-DNB-CM-2017-0062). Specimens are deposited in the herpetological collection of the Instituto Nacional de Biodiversidad (Quito, Ecuador).

Figure 1 (A) Map of Ecuador showing the distribution ranges of Nymphargus laurae: type locality (red star) and new locality (specimens INABIO15383-84; red dot), and phylogenetic sister species of N. laurae: N. siren (yellow dots) and N. humboldti (green dots). Interlined rectangle in A delimit the area shown in B. (B) Environmental risk surface (Ortega-Andrade et al., 2021) and protected areas in the distribution range of N. laurae. Numbers correspond to the following protected areas: (1) Cayambe-Coca National Park, (2) Sumaco-Napo-Galeras National Park, (3) Antisana Ecological Reserve, (4) Colonso-Chalupas Biological Reserve, (5) Llanganates National Park. Note high risk modelled for the type locality, which is excluded from the National System of Protected Areas of Ecuador. (C) Satellite image (2019, Google Earth) of the Upper Rio Napo valley, type locality near the town of Loreto (ca. 0.666670°S, 77.316700°W, ca. 500 m elevation), slopes of the Sumaco Volcano, on the Cordillera Oriental, eastern slopes of the Andes, Provincia de Orellana, República del Ecuador.

Morphological characteristics

Terminology, characters and measurements follow formats and definitions described by Cisneros-Heredia & McDiarmid (2007), Watters et al. (2016) and Guayasamin et al. (2020). Examined frogs were photographed in life, anaesthetised with lidocaine 2%; specimens and tissues were fixed in 96% ethanol and preserved separately in 75% ethanol. Sex and maturity were determined by directly examining gonads through dissections and noting secondary sexual characters (i.e., vocal slits and nuptial pads). Colour patterns are based on photographs and annotations of living specimens taken in the field. Adjective “enamelled” is used to describe the shiny white colouration produced by the accumulation of iridophores (Lynch & Duellman, 1973; Cisneros-Heredia & McDiarmid, 2007).

The following measurements were taken with a digital calliper (0.02 mm accuracy, rounded to nearest 0.1 mm) under a stereomicroscope: snout-vent length (SVL); head width (HW); head length (HL), snout length (SL); interorbital distance (IOD), horizontal eye diameter (ED), internarial distance (IND), eye-nostril distance (EN), horizontal tympanum diameter (TD); upper eyelid width (UEW), forearm length (FLL), hand length (HAL), finger IV disk width (Fin4DW), thigh length (THL), tibia length (TL), foot length (FL).

Morphological data used for comparisons were obtained from direct examination of specimens deposited in the following collections: The Natural History Museum, Department of Zoology, London (BMNH); División de Herpetología, Instituto Nacional de Biodiversidad, Quito (DHMECN); Instituto de Ciencias Naturales, Universidad Nacional de Colombia, Bogotá (ICN); University of Kansas Natural History Museum, Lawrence, KS (KU); Museum of Comparative Zoology, Harvard University, Cambridge, MA (MCZ); Museo de Zoología, Pontificia Universidad Católica del Ecuador, Quito (QCAZ); National Museum of Natural History, Smithsonian Institution, Washington, D.C. (USNM); Museo de Zoología, Universidad San Francisco de Quito, Quito (ZSFQ).

DNA extraction, amplification, sequencing, and phylogenetic analyses

Genomic DNA was extracted from hepatic tissue preserved in 96% ethanol from specimen DHMECN 15383, using the “Isolation of Genomic DNA” protocol, Wizard Genomic DNA Purification Kit (Promega, 2019). Three mitochondrial gene fragments [12S ribosomal rRNA gene, 16S ribosomal rRNA gene, Cytochrome C Oxidase Subunit 1 (COI)], and one nuclear gene [Recombination Activating Protein 1 gene (RAG1)] were amplified using the Polymerase Chain Reaction (Saiki et al., 1988). Primers for 12S gene (12S H10-FWR 5′-CACYTTCCRGTRCRYTTACCRTGTTACGACTT-3′/12S L4E-REV 5′-TACACATGCAAGTYTCCGC-3′), 16S gene (16Sar-L-FWR 5′-CGCCTGTTTATCAAAAACAT-3′/16Sbr-H-REV 5′-CCGGTCTGAACTCAGATCACGT-3′), COI (dgLCO-1490-FWR 5′-GGTCAACAAATCATAAAGAYATYGG-3′/dgHCO-2198-REV 5′-TAAACTTCAGGGTGACCAAARAAYCA-3′) and RAG1 (RAG1-R182 5′GCCATAACTGCTGGAGCATYAT3′/RAG1-R270 5′AGYAGATGTTGCCTGGGTCTTC3′), were used (Heinicke, Duellman & Hedges, 2007). Each PCR reaction was composed of a 25 μl reaction mix containing: 12.5 μl GoTaq Green Master Mix, 4.5 μl H20, 1 μl on 10 μM of Forward primer and 1 μl on 10 μM Reverse primer, and 5 μl of genomic DNA. We perform amplifications on an Applied Biosystems GeneAmp PCR System 9,700 thermal cycler. The amplification program for 12S primers was set with an initial denaturation of 94 °C (5 min) followed by 38 cycles of 94 °C (30 s), 49 °C (30 s), 72 °C (1 min), with a final extension temperature of 72 °C (7 min) and 4 °C for an unlimited period; for 16S primers was set with an initial denaturation of 95 °C (5 min) followed by 30 cycles of 95 °C (30 s), 57 °C (30 s), 72 °C (45 s), with a final extension temperature of 72 °C (5 min) and 4 °C for an unlimited period; for COI primers was set with an initial denaturation of 95 °C (5 min) followed by 30 cycles of 95 °C (30 s), 48 °C (30 s), 72 °C (45 s), with a final extension temperature of 72 °C (5 min) and 4 °C for an unlimited period; and for RAG1 primers was set with an initial denaturation of 94 °C (5 min) followed by 38 cycles of 94 °C (30 s), 52 °C (30 s), 72 °C (1 min), with a final extension temperature of 72 °C (7 min) and 4 °C for an unlimited period. Amplified DNA products were visualised by electrophoresis on a 2% agarose gel and post-staining with Tris/Borate/EDTA buffer (TBE) under blue light. PCR-amplified products were purified using Illustra™ ExoProStar™ Enzymatic PCR and Sequencing Clean-Up Kit. Sequencing was performed in both DNA strain directions and undertaken by Macrogen, Seoul, South Korea (http://www.macrogen.com). Chromatographs resulting from sequencing were revised and edited using Geneious Prime v.2020.0.5 software (Kearse et al., 2012). New sequences were deposited in GenBank with the following accession numbers: (12S: MZ820691; 16S: MZ831508; COI: MZ828399; RAG1: MZ835991).

To infer the phylogenetic position of Nymphargus laurae, we included sequences for 119 species (Table S1) selected from Guayasamin et al. (2020), obtained from the NCBI GenBank database (National Center for Biotechnology Information NCBI, 2020). We included two species for Allophrynidae, a sister lineage of Centrolenidae, as outgroups (Table S1) and rooted the phylogeny with Allophryne ruthveni. Alignments were reviewed and edited manually to remove regions with a high proportion of missing data at the edges and hypervariable regions with Geneious Prime v.2020.0.5 software (Kearse et al., 2012). Highly variable regions into the alignments are subject to the accumulation of gaps caused by deletions, insertions and substitution mutation, commonly identified to contributed to the inaccuracy of phylogenetic inference (Dwivedi & Gadagkar, 2009). We used Mesquite v3.61 software (Maddison & Maddison, 2019) to store the sequences and to create a concatenated matrix of all genes (12S, 16S, COI and RAG1). Because our combined data set comprised two ribosomal genes (12S and 16S), one protein-coding mitochondrial gene (COI) and one nuclear gene (RAG1), we tested a matrix with 8 partitions with PartitionFinder2 v2.1.1 (Lanfear et al., 2012), in the CIPRES Science Gateway V.3.3 (Miller, Pfeiffer & Schwartz, 2010). This process resulted in the selection of GTR + I + G as the optimal model for nucleotide substitution for all partitions.

Phylogenetic analyses were conducted using Maximum Likelihood (ML) and Bayesian Methods (BA) on the aligned matrix in the CIPRES Science Gateway V.3.3. ML analyses were performed using Garli v2.0 [Genetic Algorithm for Rapid Likelihood Inference; (Zwickl, 2006)]. We perform a total of 10 runs to reduce the probability of inferring a suboptimal solution; all the other settings were set on default. Node support was evaluated using 1,000 bootstrap pseudoreplicates. Bayesian phylogenetic analyses were performed in MrBayes v3.2.2 (Ronquist & Huelsenbeck, 2003), using five runs of the Monte Carlo Markov Chain (MCMC) algorithm for 20 million generations each, with four heated chains (0.2 heating parameter). Trees were sampled every 20,000 generations, with burning of 25% of the total trees. To evaluate the effective sampling size of the five independent, uncorrelated runs, we used the statistical number of effectively independent draws from the posterior (ESS > 200) visualised with Tracer v1.6. (Rambaut et al., 2013). Phylogenetic trees were edited using FigTree v1.4.2. (Rambaut, 2014).

Conservation status

We assessed the conservation status of Nymphargus laurae based on an environmental risk surface model (0 = no threats, 1 = maximum threat value) produced for Ecuadorian amphibians by Ortega-Andrade et al. (2021), a satellite image (2019, Google Earth) from the type locality of the species, and a map of the National System of Protected Areas (http://ide.ambiente.gob.ec/mapainteractivo/). We classify the extinction risk of N. laurae based on the categories and criteria presented by IUCN (2012, 2019).

Results

Surveys at the Colonso-Chalupas Biological Reserve resulted in the collection of two specimens of Nymphargus laurae. A subadult female (DHMECN 15383) was found at a stream near the Ikiam’s Scientific Station on 17 October 2016 by H. Mauricio Ortega-Andrade. An adult male (DHMECN 15384) was collected at Narpa stream, on 09 June 2018 by Miguel Gómez Laporta and H. Mauricio Ortega-Andrade. Both specimens were found active on leaves of riverine vegetation up to 4.5 meters, next to small creeks, at night between 22h00–23h00. Nymphargus laurae was found in syntopy with N. cochranae, Pristimantis quaquaversus, P. malli, P. incomptus, P. ventrimarmoratus, P. aff. petersi, and P. aff. conspicillatus. The female was classified as subadult by having unconvoluted oviducts and immature ovarian eggs.

Both specimens are very similar to the holotype in their anatomy and colouration, showing all diagnostic characters described for Nymphargus laurae: (1) vomerine teeth absent; (2) snout truncated in dorsal and lateral views; (3) tympanic annulus evident; (4) dorsal skin shagreen with slightly elevated warts corresponding to ocelli and scattered spicules, (5) ventral skin granular, without cloacal ornamentation except for a pair of large flat tubercles; (6) parietal peritoneum white, covering 2/3 of the abdomen; all other peritonea clear; (7) liver lobed, hepatic peritoneum clear; (8) humeral spine absent; (9) basal webbing between fingers I, II and III, outer fingers III 2 ⅔–2 ⅓ IV; (10) webbing on feet similar to holotype (see below); (11) no dermal folds or tubercles on hands, arms, feet or tarsi; (12) unpigmented nuptial pad Type-I, concealed prepollex; (13) Finger II longer than Finger I; (14) eye diameter larger than the width of disc on Finger III; (15) colouration in life, green with ocelli (yellow spots surrounded by black), and in preservative, lavender with ocelli (cream-coloured centre surrounded by dark lavender).

The new specimens show no relevant differences with the holotype, and observed differences fall within known intraspecific variation found also between specimens of other congeneric species. The male specimen (DHEMCN 15384) has SVL = 22.1 mm, slightly larger than the male holotype (USNM 288453, 19.9 mm SVL), and both are smaller than the subadult female (DHMECN 15383, 22.3 mm SVL). Differences in measurements and proportions between these males are probably due to intraspecific variation (Table 1). The male has a combination of large and small spicules (visible under magnification) on the head, dorsum, and flanks, but spicules on the lower part of dorsum and eyelids are smaller. The female has smaller spicules compared to the male on the head, dorsum and flanks. A spicule is present in the centre of each ocellus, being more prominent and pointed when compared to other body spicules. The female has the tympanic annulus proportionally more covered by the supratympanic fold than in males. Hand webbing in the new specimens (basal webbing between fingers I, II and III, outer fingers III 2 ⅔–2 ⅓ IV; Fig. 2A) is very similar to the holotype (III 2 ⅔–2 ½ IV), and feet webbing shows slight variation: I 2–2 ¾ II 1 ½–2 ¾ III 1 ½–2 ¾ IV 2 ¾–1 ½ V in the female and I 2 ¾–2 ¾ II 1 ½–2 ¾ III 1 ½–2 ¾ IV 2 ½–1 ¾ V in the male (Fig. 2B) (I 2−–2+II 1 ½–2+III 1+–2 ½ IV 2 ½–1 ½ V in the male holotype). The male (DHMECN 15384) has two papillae on discs of Toe I and II (Fig. 2C). The female (DHMECN 15383) lacks papillae on toes. The holotype of N. laurae has two papillae on each toe disc, except for Toe V.

Table 1 Morphometric measurements (mm) in specimens of Nymphargus laurae.

Character	Male (Holotype)
USNM 288453	Male	Subadult female	
INABIO15384	INABIO15383	
HW	7.4	7.8	8.0	
SVL	19.9	22.1	22.3	
TL	11.7	13.8	13.6	
IOD	3.8	3.1	2.8	
HL	6.9	6.9	6.5	
ED	2.9	3.4	3.6	
IND	1.6	2.3	2.7	
EN	1.7	2.1	3.0	
FL	8.7	10.5	10.3	
TD	–	0.9	0.7	
THL	–	12.3	11.7	
SL	–	3.1	2.8	
FLL	–	5.0	4.4	
UEW	–	2.1	2.9	
HAL	–	7.8	7.4	
Fin 4DW	–	1.5	2.0	
HW/HL	1.1	1.1	1.2	
HW/SVL	0.4	0.4	0.4	
HL/SVL	0.4	0.3	0.3	
EN/HL	0.3	0.3	0.5	
ED/HL	0.4	0.5	0.5	
IOD/ED	1.3	0.9	0.8	
EN/ED	0.6	0.6	0.8	
EN/IOD	0.5	0.7	1.1	
TL/SVL	0.6	0.6	0.6	
FL/SVL	0.4	0.5	0.5	
HAL/SVL	–	0.4	0.3	
FLL/SVL	–	0.2	0.2	
ED/Fin 4DW	–	2.3	1.8	

Figure 2 Hand (A), foot (B) and papillae (C) of Nymphargus laurae (INABIO15384). Tags and background color have been digitally removed.

In preservation, the new specimens show similar colourations to the holotype. However, the female shows a lavender dorsum, while the new male and the holotype have cream dorsum with lavender tones. The female has 19 ocelli on the body and eight on the head (Fig. 3A), and the male has six ocelli on the body and three on the head (Fig. 3B) (14 on the body and five on the head of the holotype). Upper eyelids are dark lavender. Fingers and toes lack melanophores. All ventral surfaces are cream. The parietal peritoneum and sclera are white, covering 2/3 of the abdomen; pericardium, digestive peritonea, hepatic peritoneum, and urogenital peritonea are clear.

Figure 3 Views of the body (dorsum and venter), of (A) subadult female INABIO15383 and (B) adult male INABIO15384 in specimens of Nymphargus laurae. Tags and background colour have been digitally removed.

The colouration in life of Nymphargus laurae remains known only from the brief description provided by Gustavo Orcés-Villagómez, Ecuadorian zoologists who donated the specimen to James A. Peters, USNM curator, and reported in the original description of the species: “green with yellow spots surrounded by black” (Cisneros-Heredia & McDiarmid, 2007). The new specimens allow for a complete description: Head green, darker than the body, lip greenish cream; dorsal surfaces of body, arms and legs green; ocelli on head and body having yellow spots surrounded by black; ocelli absent on arms and legs; upper flanks coloured as dorsum but lower flanks cream, with a sharp division between both; hands, finger, feet and toes yellowish-green, with yellow discs; nuptial pad cream (Fig. 4). Throat greenish-white, all other ventral surfaces cream white. Yellow circumpupillary ring and whitish iris with thin dark reticulations and dark flecks concentrated towards the middle (Fig. 4). Nictitating membrane yellowish, without reticulations. Green bones.

Figure 4 Nymphargus laurae (INABIO15383), (A) dorsal view, (B) side view, (C) front view and (D) ventral view.

We reconstructed the evolutionary tree (Fig. 5) of Nymphargus laurae with a dataset including 120 taxa and 2823 nucleotides (in the aligned matrix). ML and BA analyses are both congruent and recovered the phylogenetic position of N. laurae as sister species of N. siren and both forming a clade sister to N. humboldti. The clade N. laurae + N. siren + N. humboldti has low BA posterior probability (<0.9 node value in Fig. 5), while the ML bootstrap value has relative high support (0.7 node value in Fig. S1). This clade is closely related to N. megacheirus and N. anomalus (Fig. 5). Phylogenetic relationships among major groups to genus level are supported with high values, in Hyalinobatrachium, Centrolene, Cochranella, Espadarana, Rulyrana, Sachatamia, Teratohyla, and Vitreorana (Fig. S1).

Figure 5 Optimal maximum likelihood tree (log likelihood = −28155.635) of Nymphargus (the clade including N. laurae is highlighted by an orange rectangle) inferred from concatenated DNA sequences of fragments of the mitochondrial genes 12S, 16S, and COI, and the nuclear gen RAG1, totaling 2,823 aligned base pairs. Circles indicate significant support values for clades recovered by Bayesian (BA) and Likelihood (ML) analyses.

Nymphargus laurae is known from two localities in the province of Napo, on the north-eastern flanks of the Andes of Ecuador, at elevations between 700–1,500 m (Fig. 1). The type locality, Loreto, was originally covered by lowland evergreen forests, and it is located on the lower slopes of the Sumaco volcano, on the upper Napo valley. Satellite images (Fig. 1) show that less than 10% of the natural forests remains at the type locality. The new locality, Colonso-Chalupas, is still covered by evergreen montane forest (Fig. 1). The environmental risk surface (ERS) results in threat values from 0 (Colonso-Chalupas) to 0.37 (Loreto), due to habitat loss and fragmentation for cattle raising and agriculture, deforestation, roads, oil pipelines, and stochastic events related with explosions of the Sumaco Volcano.

Discussion

The records of Nymphargus laurae presented in this paper correspond to the first report of the species after 66 years from its original collection. The Colonso-Chalupas Biological Reserve is the second known locality of N. laurae, extending its geographic range in ca. 77 km SW from the type locality, at Loreto, province of Orellana, Ecuador (Cisneros-Heredia & McDiarmid, 2007). These records also extend the altitudinal range of the species from ca. 700 m (see comments on the elevation of Loreto by Urgilés, Sanchez-Nivicela & Cisneros-Heredia (2017) up to 1,500 m. Despite the altitudinal difference, both localities are in the same biogeographic region and watershed, and no significant biogeographic barriers separate them. Nymphargus laurae maybe more widespread than currently known, but possibly it is endemic to north-eastern Ecuador.

Dorsal colouration pattern showing ocelli with yellow centre surrounded by black on a green dorsum is shared by three ocellated glassfrogs: N. cochranae, N. laurae and N. lindae. These species share a common biogeographic pattern across the eastern Andean slopes in Ecuador, with N. cochranae being widespread across the eastern Andean slopes of the Andes of Ecuador and southern Colombia and sympatric with N. laurae (Cisneros-Heredia & McDiarmid, 2005; 2006; 2007, this paper). All three species are very similar in their morphology and colouration, and when a single known specimen was available for N. laurae, the differentiation between this species and N. lindae was weak and it was suggested that N. cochranae and N. laurae may be synonyms (Guayasamin et al., 2020). Now we can provide strong evidence for the distinctiveness of N. laurae, which is not closely related to N. cochranae nor N. lindae, based on morphological, chromatic, and molecular data. Externally, N. laurae differs from N. cochranae by having much larger ocelli (ocelli in N. cochranae are small, and in some specimens they are so small that without close inspection, they appear to be just dark spots); ocelli with yellow centre (orange centre in N. cochranae), Finger II longer than Finger I (Figer I > Finger II in N. cochranae); distal subarticular tubercle of fourth finger bifurcate; indistinct outer metatarsal tubercle; supernumerary tubercles present; no ocelli on forearms and shanks (present in some N. cochranae); no vomerine teeth (present in some N. cochranae); and smaller body size (23.8–31.6 mm SVL in males of N. cochranae vs. 19.9–22.3 mm SVL in males of N. laurae). Nymphargus laurae and N. lindae are very similar, but N. lindae is diagnosable due to the present of vomerine teeth (absent in N. laurae), low ulnar and tarsal folds present (absent in N. laurae), and slightly larger body size (19.9–22.3 mm SVL in males of N. laurae vs. 23.0–26.5 mm SVL in males of N. lindae). The condition of the papillae at the tip of toes was used as a diagnostic character in the original description of N. lindae, but it is not a useful taxonomic character due to its variation in N. laurae. Absence of papillae in the female and in some toes in the new male of N. laurae (Fig. 2B) suggest that papillae show intraspecific variation or is of external origin. The presence of papillae on toes discs was not used as a diagnostic character in the original description of N. laurae. Cisneros-Heredia & McDiarmid (2007) actually noted that intraspecific variation was observed in the presence/absence of papillae on toe discs of other congeneric species (i.e., Nymphargus ignotus). We have seen similar papillae in some specimens of Chimerella, Vitreorana and Espadarana, showing intraspecific variation (e.g., Espadarana prosoblepon, Vitreorana ritae).

Phylogenetic analyses place Nymphargus laurae in a clade with N. siren and N. humboldti (Fig. 5). These results are interesting due to the colouration differences among N. laurae, N. humboldti and N. siren and their close distribution in nearby areas at the Sumaco volcano and the Guacamayos mountain range. Nymphargus siren and N. humboldti are almost identical, the only phenotypic diagnostic character being the smaller body size of N. siren. However, our phylogenetic information shows that, despite their similarities, they are not sister to each other. Nymphargus siren is distributed on the eastern Andean slopes from southern Colombia to northern Ecuador, at elevations between 1,410–2,000 m; N. humboldti, is known from two localities on the eastern Andean slopes of central Ecuador, at elevations between 1,770–2,400 m (Guayasamin et al., 2020); and N. laurae is restricted to lowland and foothill forests along the Upper Napo River basin (Cisneros-Heredia & McDiarmid, 2007, this paper). Our results suggest the dispersal of this clade occurred in the northern Andes, along montane forest in the upper Napo River basin, Guacamayos mountain range and Sumaco volcano (Fig. 1A). Although these species have similar elevations and distributional ranges, the role of morphological, behavioural, bioacoustics and physiological features (i.e., climatic tolerances) is still intriguing, regarding their evolution and biogeographical diversification in eastern Andes of Ecuador.

Based on data provided herein, we propose the following extinction risk assessment for Nymphargus laurae: (1) N. laurae has suffered population reductions, based on the continuous decline in habitat quality at its type locality and surroundings, where no recent record for the species have been obtained despite surveys. Habitat quality at Colonso-Chalupas is better by being part of a protected area. However, since only three specimens are known for the species, we refrain from using criterion A until more data are available to at least inferred the population status of the species; (2) the species is known from just two localities with different conservation conditions, thus each one should be evaluated as a different threat-defined location; (3) an EOO cannot be estimated with two localities but the estimated AOO is 8 km2, which is within the threshold for Critically Endangered (<10 km2). However, we consider that it is possible that the geographic range of N. laurae is larger, closer to the threshold for Endangered (10–500 km2) under criterion B2; (4) the type locality and any potential locality outside of Colonso-Chalupas are under ongoing habitat decline due to forest loss and water pollution. This information suggests the extinction risk of N. laurae is relatively high and we propose that it should be classified under the IUCN category of Endangered (EN) based on criteria B2ab (iii, iv). Although N. laurae now is expected to have a wider distribution, urgent conservation actions are encouraged for this species and other range-restricted amphibians the eastern Andes slopes of Ecuador.

Conclusions

We provide new information about Nymphargus laurae, a species previously known from a single specimen collected decades ago. Our new specimens collected at the Colonso-Chalupas Biological Reserve increase the geographic range of the species along the north-eastern slopes of the Ecuadorian Andes. New insights into the morphology, colouration, and phylogeny of N. laurae demonstrate its distinctiveness among other ocellated glassfrogs, with which it is not closely related because it is part of a clade with N. siren and N. humboldti. Although now known from a second locality, the geographic range of N. laurae is still limited and habitat loss and fragmentation are threatening the long-term survival of populations outside of protected areas, thus we suggest that the species’ extinction risk should be categorised as Endangered at the global and national level and conservation actions are urgently encouraged. The importance of research in unexplored areas must be a national priority to document the biodiversity associated, especially for range-restricted species and in little-explored protected areas.

Supplemental Information

Supplemental Information 1 Optimal maximum likelihood tree (log likelihood = −28155.635), showing the phylogenetic relationships among 119 species of Centrolenidae and two outgroup taxa.

Values above nodes are posterior probabilities resulting from Bayesian phylogenetic analyses (values < 0.9 not shown, black circles = 1). Numbers bellow nodes correspond to non-parametric bootstraps (values < 0.70 not shown, black circles = 1).

Click here for additional data file.

Supplemental Information 2 Taxa and genetic markers used in this study.

Sequences generated in previous studies were downloaded from GenBank. Newly generated sequences are in bold blue.

Click here for additional data file.

We thank Miguel Gómez-Laporta, Michelle Guachamin, and Jimmy Velasteguí for field companion and support; Andrea Carrera for provide support for the molecular labwork; and the reviewers for their comments. We are grateful to the following people and institutions who provided access to specimens: Jeff Streicher, David Gower and Mark Wilkinson (BMNH), Mario Yánez-Muñoz (DHMECN), John D. Lynch (ICN), Linda Trueb, William E. Duellman and John E. Simmons (KU), James Hanken and Jose Rosado (MCZ), Santiago Ron (QCAZ), George Zug, Roy McDiarmid and Ron Heyer (USNM), and Carolina Reyes-Puig and Emilia Peñaherrera (ZSFQ). New specimens of N. laurae were collected as part of the project “On the quest of the golden fleece in Amazonia: The first herpetological DNA—barcoding expedition to unexplored areas on the Napo watershed, Ecuador”.

Additional Information and Declarations

Competing Interests

Author Contributions

Animal Ethics

DNA Deposition

Data Availability

The authors declare that they have no competing interests.

María José Sánchez-Carvajal conceived and designed the experiments, performed the experiments, analyzed the data, prepared figures and/or tables, authored or reviewed drafts of the paper, and approved the final draft.

Grace C. Reyes-Ortega conceived and designed the experiments, analyzed the data, authored or reviewed drafts of the paper, logistics and organization of fieldwork and expeditions, and approved the final draft.

Diego F. Cisneros-Heredia conceived and designed the experiments, performed the experiments, analyzed the data, authored or reviewed drafts of the paper, and approved the final draft.

H. Mauricio Ortega-Andrade conceived and designed the experiments, performed the experiments, analyzed the data, prepared figures and/or tables, authored or reviewed drafts of the paper, logistics and organization of fieldwork and expeditions, and approved the final draft.

The following information was supplied relating to ethical approvals (i.e., approving body and any reference numbers):

Environmental Ministry of Ecuador provided full approval for this research (MAE-DNB-CM-2017-0062).

The following information was supplied regarding the deposition of DNA sequences:

The new sequences are available in GenBank: 12S MZ820691; 16S MZ831508; COI MZ828399; RAG1 MZ835991.

The following information was supplied regarding data availability:

The raw data is available in the Supplemental Files.

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
