# Peer review of "Rediscovery of Laura’s glassfrog Nymphargus laurae (Anura: Centrolenidae) with new data on its morphology, colouration, phylogenetic position and conservation in Ecuador"

_PeerJ, doi:10.7717/peerj.12644_

## Round 0.1 · original submission · Minor Revisions

Please address comments by both reviewers when preparing the revised version of your manuscript.

Reviewer 1 ·

Basic reporting

The article is clearly written, well-structured and uses appropriate technical language. The introduction provides sufficient background on the problem and cites all relevant previous literature. Methods, results, discussion, figures, tables, and raw data shared conforms all the standards of PeerJ. However, there are some minor issues, namely:

In the methods section, it is unclear why the authors decided to exclude hypervariable regions from the alignments. I suggest including a brief justification for this action (see comment on line 151 in the annotated pdf).

Also in methods, I consider that the part corresponding to the evaluation of the best partition schemes and replacement models using PartitionFinder should be better explained. As currently described, it is unclear how many and which partitions were selected as part of the optimal scheme. In addition, I think the claim that the same replacement model was selected for all partitions should be verified. I included detailed comments on this topic in lines 157-158 of the annotated pdf.

Several species names must be corrected in the figures 5 and S1, and in the Table S1 (see notes on Fig. 5 in the annotated pdf and in Table S1, annexed at the end of the annotated pdf).

A couple of citations are missing in the references (see lines 57 and 268 in the annotated pdf) and all the citations corresponding to the primers used in the study are also missing (see lines 126-131 in the annotated pdf). Other minor details that need to be corrected are highlighted throughout the annotated pdf.

Experimental design

No comment.

Validity of the findings

No comment.

Annotated reviews are not available for download in order to protect the identity of reviewers who chose to remain anonymous.

Reviewer 2 ·

Basic reporting

The structure of the document is clear, concise, and includes all the elements required to comprehend the results obtained and their relevance.
The document includes the basic literature relevant to the study field. The supporting data are effectively provided by the authors. The colors of N laurae and N siren are inverted in the legend of figure 1 so it has to be corrected.
The names of the species in the phylogenetic tree (Figure 5) must be italicized
Line 179: "presented by IUCN (2012, 2019)"

Experimental design

Please provide in the Methods section the information about the museum where the specimens are deposited. Some information is available in the results section but it has to be specified before.

Validity of the findings

The phylogenetic position as well as the morphological comparisons with similar and close-related species are valuable results on this research.

The main concern about the findings is the use of only one specimen on the molecular analysis, the geographic distance between the type locality and this report, and the morphological variation on diagnosis characters (toe papillae) mentioned by the authors. Therefore, the authors must provide enough information to 1) confirm that both individuals collected are from the same species and 2) the species is not a new fourth ocellated Nymphargus species.

The authors must justify the exclusion of hypervariable regions (I suppose are those from ribosomal sequences) in the analysis.

Additional comments

I wish to congratulate the authors on their high-quality work.

---

## Round 0.2 · accepted · Accept

You have provided an improved version of their previous manuscript, which I find acceptable for publication. Thanks for your work!